# The Notorious Soilborne Pathogenic Fungus *Sclerotinia sclerotiorum*: An Update on Genes Studied with Mutant Analysis

**DOI:** 10.3390/pathogens9010027

**Published:** 2019-12-27

**Authors:** Shitou Xia, Yan Xu, Ryan Hoy, Julia Zhang, Lei Qin, Xin Li

**Affiliations:** 1Hunan Provincial Key Laboratory of Phytohormones and Growth Development, Hunan Agricultural University, Changsha 410128, China; leiqin@stu.hunau.edu.cn; 2Michael Smith Laboratories, University of British Columbia, Vancouver, BC V6T 1Z4, Canada; yan.xu@msl.ubc.ca (Y.X.); rhoy@alumni.ubc.ca (R.H.); 3Department of Botany, University of British Columbia, Vancouver, BC V6T 1Z4, Canada; 4School of Kinesiology, University of British Columbia, Vancouver, BC V6T 1Z1, Canada; Julia.Zhang@hli.ubc.ca

**Keywords:** soilborne fungal pathogen, *Sclerotinia sclerotiorum*, fungal pathogenesis, sclerotial formation, apothecial development, fungal growth

## Abstract

Ascomycete *Sclerotinia sclerotiorum* (Lib.) de Bary is one of the most damaging soilborne fungal pathogens affecting hundreds of plant hosts, including many economically important crops. Its genomic sequence has been available for less than a decade, and it was recently updated with higher completion and better gene annotation. Here, we review key molecular findings on the unique biology and pathogenesis process of *S. sclerotiorum*, focusing on genes that have been studied in depth using mutant analysis. Analyses of these genes have revealed critical players in the basic biological processes of this unique pathogen, including mycelial growth, appressorium establishment, sclerotial formation, apothecial and ascospore development, and virulence. Additionally, the synthesis has uncovered gaps in the current knowledge regarding this fungus. We hope that this review will serve to build a better current understanding of the biology of this under-studied notorious soilborne pathogenic fungus.

## 1. Introduction 

Ascomycete *Sclerotinia sclerotiorum* (Lib.) de Bary is a soilborne pathogenic fungus, which was first described as *Peziza sclerotiorum* by Libert [1]. In 1945, Whetzel [2] established *S. sclerotiorum* (Lib.) de Bary as the major species of *Sclerotinia*, as it is the best-known and most economically important pathogen of the genus. It was proposed as the representative species for the genus *Sclerotinia* in the 1970s [3,4]. The Special Committee on Fungi and Lichens recommended the generic name *Sclerotinia*, and so did the General Committee of the International Association of Plant Taxonomists later on [5].

*S. sclerotiorum* belongs to the family *Sclerotiniaceae* [2] of the class Leotiomycetes. *Sclerotiniaceae* also includes two other very closely related species, *S. minor* and *S. trifoliorum*. All three species produce vegetative resting structures, termed sclerotia, and mushroom-like reproductive structures, named apothecia, during their life cycles. However, *S. minor* forms small sesame-sized sclerotia, whereas *S. sclerotiorum* and *S. trifoliorum* can form large sclerotia as big as peas. They all lack an obvious conidial stage. *S. sclerotiorum* has the broadest host range, which includes both herbaceous and succulent plants, and even some woody ornamentals and monocots. *S. trifoliorum* was isolated from vegetable legumes, while *S. minor* has been found on sunflowers, tomatoes, carrots, peanuts, and lettuce. 

*S. sclerotiorum* is a highly damaging pathogen with diverse infection modes and a double feeding lifestyle of both biotroph and necrotroph (Figure 1) [6,7]. It attacks host plants either by means of ascospores that can be discharged forcibly upwards from apothecia into the air, or by mycelium arising from infected tissue or from germinated sclerotia [8]. When ascospores land on susceptible host tissue, they can germinate under favorable conditions and start a new cycle of infection. To date, the microconidia of *S. sclerotiorum* have been observed amongst the mycelium and on the surfaces of sclerotia. However, no evidence of their functionality has been identified. Under moist and cool conditions, this fungus rapidly grows inside the infected host tissues and develops symptoms of browning, water-soaking, and a white, cotton-like mycelium, which leads to necrosis, stunting, premature ripening, and wilting of the host [8]. Therefore, the diseases that it causes have been given names, including stem rot, drop, crown rot, cottony rot, watery soft rot, blossom blight, and Sclerotinia blight. Upon killing the host, the fungus can saprophytically grow on the dead plant tissue. Sclerotia are later abundantly formed on the host surface and cavities, in plant debris, and in soil, where they are able to remain dormant for up to 10 years [9].

*S. sclerotiorum* has an extremely broad host range, which consists of more than 600 plant species, including almost all dicotyledonous and some monocotyledonous plants [10,11]. It can infect many economically important crops, such as canola; legumes such as soybean and peanut; sunflower; various vegetables such as lettuce and tomato; and monocotyledonous plants such as tulip and onion [10,12]. It is therefore not surprising that it causes significant economic losses globally every year [13]. For example, in China, oilseed rape yield losses that are caused by SSR (Sclerotinia stem rot) usually range from 10 to 20% and they may be up to 80% for severe SSR outbreak seasons [14,15,16]. In The United States, annual losses that are caused by this pathogen have exceeded $200 million [12]. 

As with most other fungal pathogens, chemical control is the most commonly used method for controlling the diseases that are caused by *S. sclerotiorum*, although it is environmentally unfriendly and can induce resistance from the pathogen. Chemical control also becomes less effective once the sclerotia are formed. Obviously, the incidence of disease would be greatly reduced if sclerotia could be either prevented from forming or destroyed in soil and plant debris. Flooding and crop rotation are good options that accelerate the decay of sclerotia [17]. However, flooding is impractical in most areas, and crop rotation has been found to be less effective as a control measure [18,19], because of the long survival periods of sclerotia in soil. Measures, like widening row spacing, and using wire trellis supports to raise foliage from the ground have been reported to be helpful [18,20]. However, factors, such as type of crop, method of cultivation, and environmental conditions, always affect the effectiveness of each control method.

For major crop diseases, the most welcoming and environmentally friendly control method is through the use of resistant cultivars. However, strong host single-gene resistance has not been found against *S. sclerotiorum* [12], which makes it difficult to improve resistance using classical breeding methods [7]. Even with mild resistance, the traits seem to be multigenic. Future development of integrated approaches may help to control the spread of the fungus with better understanding of the biology of this notorious pathogen and better QTL mapping strategies.

In this review, we focus on recent key molecular findings on the unique biology and pathogenesis process of *S. sclerotiorum*. A brief summary of the genome of the pathogen will also be discussed. The genomic sequences of *S. sclerotiorum* have been available for less than a decade [21]. However, it was recently updated with higher completion and better gene annotation [22]. We hope that this review will serve to build a better current understanding of the biology of this notorious soilborne pathogenic fungus.

## 2. The Features of the *S. sclerotiorum* Genome

### 2.1. Genome Sequences

The genome of *S. sclerotiorum* strain 1980, originally isolated from beans in western Nebraska USA, was first sequenced in 2011 using Sanger technology with 9.1x coverage. As approximately 1.6 Mb of sequences were not covered by the 38.0 Mb scaffolds, and thus missed from the assembly, the final size of *S. sclerotiorum* genome was estimated to be approximately 38.3 Mb, which consists of 16 linkage groups likely corresponding to 16 chromosomes (GenBank accession numbers: AAGT01000000) [21]. Most of the uncovered regions are located in the middle of chromosomes, which probably correspond to centromeres. The genome was predicted to contain 14,522 genes, with average GC contents of 41.8%. After removing genes encoding small proteins that are less than 100 amino acids in length or without evidence of expression (from ESTs and/or microarray signals), 11,860 predicted proteins with high confidence were deduced [21]. 

Initial analyses of the genome sequences revealed a large number of virulence related genes that were possessed by the *S. sclerotiorum* 1980 strain, including genes encoding cell wall degrading enzymes (CWDEs) and biosynthesis genes of phytotoxins and other secondary metabolites [21]. Amselem et al. also identified a significant amount of repetitive transposable elements (TEs), which comprises approximately 7.7% of the whole genome. This led to a postulation that the *S. sclerotiorum* genome experienced a recent major remodeling that was associated with a dramatic expansion of TEs. The TE expansion in *S. sclerotiorum* might also have an impact on its genome organization and function of gene inactivation, modification, or expression regulation [21]. 

Derbyshire et al. recently sequenced the genome sequence of the same *S. sclerotiorum* 1980 strain to near completion in 2017 using PacBio technology. It was annotated using extensive RNA-Seq data and manual curation. Therefore, this version of the whole-genome sequence became the choice of standard for *Sclerotinia*. They identified 70 candidate effector genes, and found a significant association between the positions of these secreted proteins and regions with a high relative RIP (repeat-induced point mutation) index, which suggested that *S. sclerotiorum* exhibits a subtly enhanced mutation rate of secreted proteins in specific genomic compartments as a result of transposition and RIP activity [23].

More recently, Derbyshire et al. sequenced the genomes of 25 field isolates of *S. sclerotiorum* that were collected from four different continents—Australia, Africa (north and south), Europe, and North America (Canada and the Northern United States). They conducted SNP-based analyses on population structure and selective sweeps [24,25]. These 25 isolates can be grouped into two major populations, where population 1 consists of 11 isolates from Canada, the USA, and France, and population 2 includes nine isolates from Australia and one from Morocco. A single candidate selective sweep was identified in the Australian and Moroccan group, which covers less than 0.001% of the genome. It is reasonable to speculate that a slow evolution rate and extensive self-fertilization of *S. sclerotiorum* may have led to a striking absence of strong selective sweeps, as there is only one gene within this 10 Kb region, which encodes a major facilitator superfamily transporter that was only negligibly expressed at late stages of *Brassica napus* infection [22]. In support of such genome conservation, *S. sclerotiorum* strains from even a tropical and a temperate region, although segregated in the populations, still exhibit similar genetic structure [26].

### 2.2. Transcriptomic and Secretomic Analysis

The availability of the high-quality genome sequences facilitated comparative transcriptomic and secretomic analysis of this notorious pathogen, which in turn facilitated biological investigations. Seifbarghi et al. (2017) was able to delineate gene expression patterns that signified transitions between pathogenic phases of *S. sclerotiorum*, namely, host penetration, ramification, and necrotrophic stages, through RNA-Seq analysis focusing on events occurring through the early (1 h) to the middle (48 h) stages of infection. These expression data provide evidence for the occurrence of a brief biotrophic phase soon after host penetration [27]. 

432 proteins were identified in *S. sclerotiorum* in an interspecies comparative analysis of the predicted secretomes of *S. sclerotiorum* and *Botrytis cinerea*. Among them, 16% of the encoding genes reside in small gene clusters that are distributed over 13 of the 16 predicted chromosomes [28]. These candidate genes provide a reservoir for future reverse genetics analyses to delineate the pathogenesis mechanisms in the *S. sclerotiorum*-host plant interactions.

Mutant analysis is paramount to establish a causal relationship between a gene and a studied biological process, as omics experiments only provide vague association clues. Hence, the rest of this review will summarize all of the genes studied in *S. sclerotiorum* so far using such convincing mutant analysis methods. A Venn diagram of the functions of these encoded proteins is provided for readers’ overview (Figure 2). We also constructed a chromosomal map for these genes to detect possible clustering events that are often associated with virulence factors (Figure 3). However, little clustering is observed, except for genes contributing to mating and melanisation, reflecting the under-studied nature of this fungus. 

## 3. Molecular Dissection of *S. sclerotiorum* Biology

Here we adopt the gene/mutant/protein nomenclature that is most commonly used by *Sclerotinia* researchers, as there are many discrepancies in *S. sclerotiorum* literature. As an example, Abc1 protein is encoded by wild-type gene that is italicized as *Abc1*. Mutant is denoted as *abc1*. Exceptions are specifically explained. Another consideration to keep in mind is the presence of huge variations in the field isolates and local hosts used in these studies (Table 1), which likely contribute to some of the discrepancies that were observed in the phenotypes of mutants of the same genes.

### 3.1. Regulation of Mycelial Growth and Virulence

During the last two decades, the availability of whole genome sequences and improved transformation as well as knockdown/knockout methods have greatly facilitated the molecular study of *S. sclerotiorum*. Most of the genes studied so far that affect growth are relatively conserved genes that also affect differentiation, such as sclerotial and apothecial development (Figure 2), which will be discussed in the next sections. In this part, we specifically discuss genes only affecting hyphal growth rate and virulence, but not differentiation, during development, as mycelial growth precedes processes of virulence and differentiation. 

It is not surprising that the mutants here with major growth defects often exhibit virulence attenuation, although there are certainly exceptions such as *mat1-2-1*, which exhibits slower growth, aberrant apothecial morphogenesis, but normal virulence [29] (discussed more later). 

The RNAi knockdown of highly conserved transcription factors *Ams2* or *Mads* lead to reduced mycelial growth rate and virulence, which suggested that they are early growth regulators. Ams2 is a GATA-box domain transcription factor. Ams2 is required for proper expression of histone and cell cycle related genes, similar as in *Schizosaccharomyces pombe* [55]. Mads is orthologous to yeast Mcm1 with a MADS-box domain, which is essential for viability, cell cycle, mating, mini-chromosome maintenance, recombination, and stress tolerance [61]. Although mutants of these early growth factors severely affect growth, they can still form sclerotia, which indicated that the later stages of fungal differentiation pathways, such as sclerotial formation and apothecial development, might be less dependent on these early growth factors. 

BAX inhibitor-1 (Bi1) is a highly conserved protein that is found in both eukaryotes and prokaryotes, exhibiting apoptosis-inhibiting activity. RNAi knockdown mutants in *S. sclerotiorum* showed more aerial hyphal growth and reduced virulence [48]. The mutants also exhibit reduced tolerance to chemical stress, as its gene expression is induced by various stresses. Bi1 is likely a stress specific regulator dedicated to both biotic and abiotic stress responses, due to its minor roles in growth and development.

Unlike the previously described genes, *PemG1* serves an opposite function in regards to growth and virulence. The RNAi mutants of *PemG1* exhibit faster mycelial growth and enhanced virulence, indicating that it is a negative regulator in these processes. Consistently, more infection cushions, more CWDE activities, and less susceptibility to salt stress were observed [54]. However, the exact mechanism on how this fungal specific protein negatively regulates growth and virulence is unclear.

### 3.2. Signaling Events Leading to Sclerotial Formation

Sclerotia of *S. sclerotiorum* initiate with small primordia and rapidly develop into white compact hyphal masses. They mature with dehydration and pigmentation once they cease growth. The melanised outer surface of the sclerotium is mostly responsible for resisting adverse conditions [8]. We will first discuss the genes affecting both growth and sclerotial formation, which are likely very upstream components that are shared by both processes. As expected for broad regulators, these mutants show not only defective growth and sclerotial malformation, but also almost always loss of virulence and other pleiotropic defects (Table 1). 

The melanisation of sclerotia is an important process for the long-term survival of *Sclerotinia* species in nature. The deletion of melanin biosynthesis genes scytalone dehydratase (*Scd1*) or trihydroxynaphthalene reductase (*Thr1*) resulted in slower mycelial growth, reduced number, size, and pigmentation of sclerotia [39], which suggested that melanisation is critical for sclerotial maturation and perhaps feedback promotion of growth. However, unlike in other pathogenic fungi, the virulence of these knockout strains is not affected, suggesting that melanin does not contribute to pathogenesis in *Sclerotinia* as it does for other fungi, like *Magnaporthe* species.

Knocking out some other conserved genes in *S. sclerotiorum* can also lead to diverse defects. For example, the peroxisomal carnitine acetyl transferase (CAT) encoding gene in *S. sclerotiorum* is *Pth2*. The CAT activity is required for acetyl-CoA transport, which is needed for fatty acid metabolism. The *pth2* knockout mutants show aberrant growth, non-sclerotial formation, and reduced oxalic acid (OA) levels and virulence [41]. In another instance, the silencing of integrin-like gene *Itl* leads to altered colony morphology, smaller and irregular shaped sclerotia, and hypovirulence [62]. Although *Cvnh* homologs seem to only exist in *Sclerotinia* and *Botrytis*, the CVNH domain is a conserved domain. The silencing of this gene lead to reduced growth rate, non-sclerotial formation and lower virulence [45].

Phosphorylation and de-phosphorylation events are critical for almost all signal transductions. Therefore, it is not surprising that a number of kinases and phosphatases are involved in both growth and sclerotial formation. Mutation of two-component histidine kinase *Shk1* leads to slower and altered hyphal growth and failed sclerotial formation [35]. Although the mutant is also sensitive to osmotic stress and shows heightened resistance to fungicides, it exhibits normal virulence. Shk1 likely works upstream of the MAPK cascade to control these processes. Smk1 and Smk3 are two MAPK kinases that have been studied in *S. sclerotiorum* so far. Mutants of these two kinases show similar growth defects and non-sclerotial formation [60,71], suggesting that they may function downstream of Shk1. 

cAMP could act upstream of the MAPKs, as exogenous application of cAMP inhibits *Smk1* expression and MAPK phosphorylation [71]. Type 2A phosphoprotein phosphatase Pph1 possibly functions downstream of the MAPK. The mutants of *pph1* show arrested growth [67]. Consistently, PP2A R2 B regulatory subunit *rgb1* mutants show aberrant sclerotial formation and failed penetration during infection [67]. 

Reactive oxygen species (ROS) play diverse roles in development and virulence in fungi. This is supported by the mutant analysis of type I catalase *Scat1*. The deletion of *Scat1* lead to slower radial growth, higher number of small sclerotia that are not properly melanised, and hypovirulence [44]. Calcium is another common factor that is involved in signaling. The study of Cna1 and Caf1 supports its contribution in *S. sclerotiorum* biology. Calcineurin Ser/Thr protein phosphatase Cna1 is conserved in eukaryotes, whose activity is dependent on calcium and calmodulin. Antisense *cna1* mutants show altered growth, sclerotial development, and hypovirulence [32]. Caf1 is a secreted protein with a putative Ca^2+^-binding EF-hand motif, which seems to be conserved in fungi. *caf1* T-DNA mutants exhibit slightly slower growth, deformed sclerotia, and hypovirulence [43]. 

Conserved transcription factors are another large group of genes that are responsible for signaling. Sfh1 is a GATA transcription factor, but with an SNF5 domain. It is a member of the housekeeping RSC (Remodels Structure of Chromatin) complex, which is an ATP-dependent chromatin remodeler that is essential for cell cycle [34]. Its diverse functions explain its involvement in growth, sclerotial development, and virulence. Transcription factor Ste12 likely acts downstream of the MAP kinase cascades. In *ste12* RNAi lines, slow growth, smaller sclerotia, and fewer appressoria, leading to hypovirulence, were observed [13]. Forkhead (FKH) box (FOX) family transcription factors are important regulators of primary metabolism, cell cycle, and morphogenesis in animals and fungi. *fkh1* RNAi mutants show slower growth, non-sclerotial formation, hypovirulence, and defective stress responses [52], suggesting that it is a transcriptional regulator that is shared by multiple pathways. 

CWDEs have been speculated to play key roles in fungal pathogenesis. In support of this, the deletion of *Xyl1* that encoded an endo-β-1, 4-xylanase leads to a loss of pathogenicity. Interesting, growth retardation, less sclerotial formation, and dense hyphal branching were also observed in the mutant [69], suggestive of a role of xylanase in development as well.

Opsins are conserved light-sensitive proteins involved in circadian rhythms. *sop1* RNAi lines show reduced growth, non-sclerotial formation, hypovirulence, and heightened sensitivities to osmotic and fungicide stresses [33]. These observations suggest that light may have diverse roles in various processes in *Sclerotinia* through opsins.

Rearrangement hotspot (Rhs) repeat-containing proteins are widely distributed in bacteria and eukaryotes. Rhs1 is a secreted protein and its orthologs seem to only be present in *Sclerotinia* and *Botrytis* species. *Rhs1* RNAi mutants show slightly slower colony growth, fewer and larger sclerotial formation, and hypovirulence [51]. These data suggest that Rhs1is likely a key virulence factor for *Sclerotinia* and it also slightly contributes to growth and sclerotial development. Cerato-platanins are fungal secreted elicitors that can trigger plant immunity and cell death. Sm1 can cause host hypersensitive response. Interestingly, besides virulence, RNAi mutants of *Sm1* also exhibit slower growth, impaired sclerotial development, and heightened sensitivities to different stresses [73], suggesting a broader role of the elicitor. However, these defects were not detected in a separate study in deletion mutant *cp1* mutating the same gene [72]. Future careful transgenic complementation experiments are needed to solve these discrepancies.

The existence of mutants that have normal mycelial growth, but do not form sclerotia, suggest that later stages of sclerotial development are separated from growth. Although most of these mutants still show attenuated virulence, some exhibit normal pathogenicity, such as *sl2* and *ggt1* (ɤ-glutamyl transpeptidase). RNAi knockdown of either *Sl2*, GAPDH (*Gpd*), or *Hex1* lead to altered sclerotial formation or melanisation, similar to its interacting proteins glyceraldehyde-3-phosphate dehydrogenase (GAPDH) and Hex [46]. However, the virulence of *sl2* is not affected, suggesting that sclerotial formation and virulence can be separately controlled by distinct downstream factors. As a fungal specific secreted protein localizing to fungal cell wall, how Sl2 affects sclerotial development is unclear. Ggt is a highly conserved enzyme that contributes to redox homeostasis. The deletion mutants of *ggt1* show much higher GSH+GSSG and hydrogen peroxide accumulation, and separation of the cortex layer of mature sclerotia from the medulla [63]. In addition, the mutants are defective in compound appressoria production, only affecting the penetration stage of infection. These mutant phenotypes suggest that GSH, GSSG, or GSH/GSSG ratio contribute to the regulation of sclerotia maturation and appressoria development.

The rest of the genes studied so far that affect sclerotial formation, but not growth, all show attenuated virulence. They likely act upstream of *Sl2* and *Ggt*. ROS increase was observed in sclerotial initials and infection cushions in *S. sclerotiorum*, likely due to the combined activities of NADPH oxidases Nox1 and Nox2. Similar to exogenous application of NADPH inhibitor or ROS scavenger, the RNAi mutants of either *Nox1* or *Nox2* exhibit failed sclerotial development [50], indicating that ROS is required for sclerotial initiation. Interestingly, reduced OA production and virulence was only observed in *nox1*, but not *nox2*, which suggests that these two genes are not fully redundant, likely due to their different expression patterns. Consistently, when the thioredoxin reductase (*Trr1*) gene that is partly responsible for ROS detoxification is silenced, less and larger sclerotia are formed [47]. However, virulence is attenuated in *Trr1* silenced strains, suggesting that an optimum ROS is needed for virulence. 

Low pH was known to promote sclerotial proliferation. The pH-sensitive transcription factor Pac1 is responsible for ambient pH-induced gene expression. Consistently, the deletion of *Pac1* caused aberrant sclerotial formation and reduced OA levels, thus leading to hypovirulence [53]. As higher levels of OA reduce pH and may affect ROS homeostasis, proper balance among ROS, OA, and pH is likely paramount for the biology of this fungal pathogen. 

### 3.3. Control of Apothecial and Ascospore Development

The apothecium consisting of a stalk (stipe) and a saucer-shaped disc is characteristic of the *Sclerotinia* genus (Figure 1). Asci develop on the upper surface of apothecia and when they mature, the ascospores are dispersed [8]. As mating-type (Mat) genes have generally been established as key regulators of sexual development of ascomycetes, it is not surprising that they affect apothecial development in the homothallic *S. sclerotiorum*. Similar to other fungi, the *S. sclerotiorum Mat* locus contains a cluster of four *Mat* genes encoding transcription factors, with *MAT1-1s* encoding an alpha-domain and *Mat1-2s* encoding high mobility group (HMG) proteins. By deletion knockout analysis [29], each of the four *Mat* genes were analyzed in regards to their contributions to regular mycelial growth, virulence, sclerotial formation, and apothecial development. While these Mats contribute little to vegetative growth, development, and pathogenesis, *Mat1-1-5*, *Mat1-1-1*, and *Mat1-2-1* are essential for early, and *Mat1-2-4* is critical for late sexual stages, including apothecial initiation and ascospore development. As transcription factors, it is also not surprising that these *Mat* gene mutants exhibit altered expression of the *Mat* locus and putative pheromone and pheromone receptor genes, explaining their potential regulatory mechanism and specific phenotypes. 

Similar to the *Mats*, another gene that specifically contributes to apothecial development is *FoxE2* [49]. FoxE2 is a member of the forkhead box (FOX) transcription factor (TF) family. FOX proteins play diverse roles in morphogenesis, development, pathogenicity, and stress responses in animals and fungi. Knockout analysis of *FoxE2* in *S. sclerotiorum* showed that it does not affect mycelial growth, sclerotial formation, or virulence. The striking mutant phenotypic similarity between the *Mats* and *FoxE2* brings the question on the relationship among these transcription factors. It could be possible that a transcriptional network that includes these transcription factors needs to be orchestrated for proper sexual stage development in *S. sclerotiorum*.

As a UV-A photoreceptor, Cry1 was suspected to be involved in apothecial development, as its expression was induced by UV during this specific developmental phase. However, no obvious defects in apothecial development were detected in the *cry1* deletion mutant [58], suggesting that it is not a major regulator, or there exists redundancy masking its effects on photomorphogenesis. 

All other mutants in the literature with defects in apothecial development show pleiotropic defects, including *sac1* (Adenylate cyclase) [70] and *nsd1* (GATA-type IVb zinc-finger transcription factor) [75], both encoding conserved proteins. cAMP is a well-known molecule with diverse roles for many biological processes in plant fungal pathogens. Although protein kinase A catalytic subunit gene (*Pka1*) was hypothesized to mediate cAMP signaling, the *pka1* single mutant exhibits non-detectable defects, likely due to genetic redundancy of *Pka2* [56]. In comparison, with dramatic cAMP level reduction, the *sac1* mutant displays slower growth, abolished pathogenicity, and failed apothecial development [70]. In other fungi, Nsd1 serves as transcription factor for development and environmental responses. In *nsd1* knockout mutants, altered hyphal growth, smaller sclerotia, failed appressorium formation leading to failed pathogenicity, and failed apothecial initiation were observed [75]. These indicate that Nsd1 is likely a general transcription factor that regulates many processes in *S. sclerotiorum*, or a specific regulator that regulates a general downstream factor that is shared by many pathways. Recently, the NO homeostasis regulating glutathione-dependent formaldehyde dehydrogenases (Fdh1) encoding gene was revealed to be one of the targets of Nsd1 [74]. The deletion mutants of *fdh1* exhibit slower mycelial growth, smaller sclerotia, and defective compound appressoria, leading to failed penetration into host tissue, being suggestive of the keys roles NO plays in these processes. The high similarity between the *fdh1* and *nsd1* mutant phenotypes indicates that *Fdh1* could be the main target of Nsd1 during growth and development.

### 3.4. Regulation of Fungal Pathogenesis of the Fungus

Following a short biotrophic phase after infection, *S. sclerotiorum* causes tissue maceration and necrosis leading to rapid cell death and host cell wall degradation during fungus colonization. Toxins and CWDEs are believed to play critical roles in promoting these processes. The study of *axp* mutants deleting a secreted arabinofuranosidase/β-xylosidase precursor reflects the importance of CWDEs in the virulence of *S. sclerotiorum*. Attenuated virulence was observed in canola. Similarly, the critical contribution of ROS in virulence is corroborated in *sod1* mutants affecting Cu/Zn superoxide dismutase [36,37]. OA is another key factor that has long been hypothesized to contribute to the pathogenicity of this pathogen. Readers are referred to two excellent reviews that have summarized the historic and comprehensive studies of the pathogenicity and virulence of *S. sclerotiorum* [65,66]. 

During infection, *S. sclerotiorum* produces large amounts of OA. Earlier physiological and pharmacological studies have highlighted its importance. In addition to virulence, UV mutants where the exact mutations were not identified could not produce OA and failed to form sclerotia [76], also suggesting a link between OA and development. It was not until the availability of genetically defined OA deficient mutants that the contribution of OA towards *Sclerotinia* biology was partly settled. In *odc2* RNAi strain where the gene encoding oxalate decarboxylase in charge of catabolism of OA is silenced, less and non-functional appressoria were formed, leading to hypovirulence [64]. These data support the involvement of developmentally regulated OA accumulation in penetration-dependent infection. In recently generated *oah* (oxaloacetate acetylhydrolase) mutants through the deletion or CRISPR, where OA biosynthesis is blocked, failed appresorium formation, hypovirulence, and delayed sclerotial formation was observed [40]. However, the deletion or T-DNA mutants of *oah* in a separate study show pH-dependent virulence alteration and normal growth and development. These discrepancies can likely be resolved through more careful examination of other mutants that are deficient in OA biosynthesis. 

Many pathogen effectors are secreted into the host to dampen host immunity and/or promote virulence. In recent years, effector biology has been focusing on the study of secreted proteins, as these effector candidates can be relatively easily identified through secretome and RNA-seq analysis. Many secreted proteins in *S. sclerotiorum* have diverse roles in both development and virulence (Table 1 and discussions in the previous sections). Examples of effectors only affecting virulence in *S. sclerotiorum* include Ssvp1, V263 and Qdo (quercetin dioxygenase). Less conserved Ssvp1 [30] and V263 [38] are both small secreted proteins that contribute to virulence. On the other hand, more conserved Qdo catalyzes the cleavage of the flavonol carbon skeleton [57], revealing the critical contributions of degrading host anti-microbial flavonol in fungal virulence. 

Besides secreted proteins, there are also non-secreted regulators that are solely dedicated to pathogen penetration and virulence regulation. The examples in *S. sclerotiorum* include Svf1 (Survival factor 1) [31], Pks13 (polyketide synthase 13) [40], and Nacα (nascent-polypeptide-associated complex alpha subunit) [59]. In yeast, Svf1 promotes survival under oxidative stress conditions. Therefore, it likely assists in coping with ROS during infection in *Sclerotinia*. Pks13 is likely involved in melanin biosynthesis in appressoria, as its CRISPR mutant exhibits albino compound appressoria without affecting virulence. Nacα is a highly conserved protein with transcriptional co-activator activity. Interesting, Nacα negatively impacts the expression of polygalacturonase-encoding genes, which likely contributes to the enhanced virulence phenotypes of the mutant. 

## 4. Summary and Future Perspectives

Over the decade, many genes that have been involved in pathogen development and pathogenesis have been characterized in *S. sclerotiorum* (Table 1). Much has been unveiled for the pathogen growth, development, and differentiation signaling pathways. Accordingly, more evidence is supporting the two-phase infection model, where the pathogen suppresses host basal defense prior to killing and degrading host cells [11]. In the short biotrophic phase, the pathogen uses well-orchestrated strategies to overcome host immunity. It might spatially achieve this via the production of OA, ROS, CWDEs, and effectors in compound appressoria or primary invasive hyphae. 

Among almost 15,000 genes encoded in its genome, less than 60 have been studied so far using careful mutant analysis. This explains the under-studied nature of this notorious pathogen and the lack of connections in the signaling pathways of its different special biological processes. Besides the traditional one-gene-at-a-time reverse genetics analysis, more robust methods, such as genome-wide CRISPR, should be developed to enhance the study of this economically important fungus. 

Due to the short biotrophic period, as well as the switching to necrotrophic phase, identifying and assigning the temporal, spatial, and tissue-specific functions to virulence factors are real challenges. Future development of cell biology and biochemical tools tailored for *S. sclerotiorum* will offer opportunities for solving many mysteries of the fungus. Considering this specific form of pathogen-host interaction system, challenges also exist ahead in applying quantitative disease resistance breeding and phenotypic screens for stage-specific defenses. More collaboration among the molecular pathologists and breeders will be critical to solve the host resistance difficulties of the pathogen. 

## Figures and Tables

**Figure 1 pathogens-09-00027-f001:**
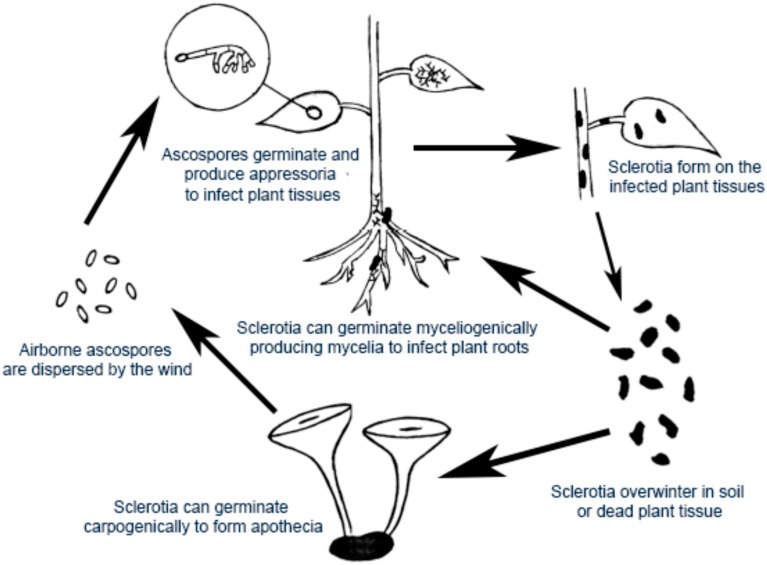
Life cycle of *Sclerotinia sclerotiorum*.

**Figure 2 pathogens-09-00027-f002:**
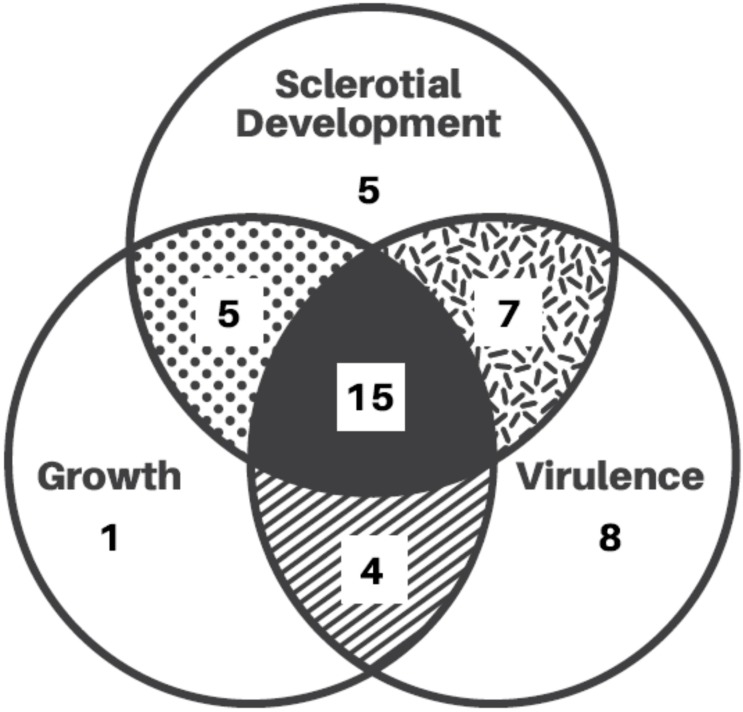
A Venn diagram summary of the genes that have been studies with mutant analysis (detailed in Table 1).

**Figure 3 pathogens-09-00027-f003:**
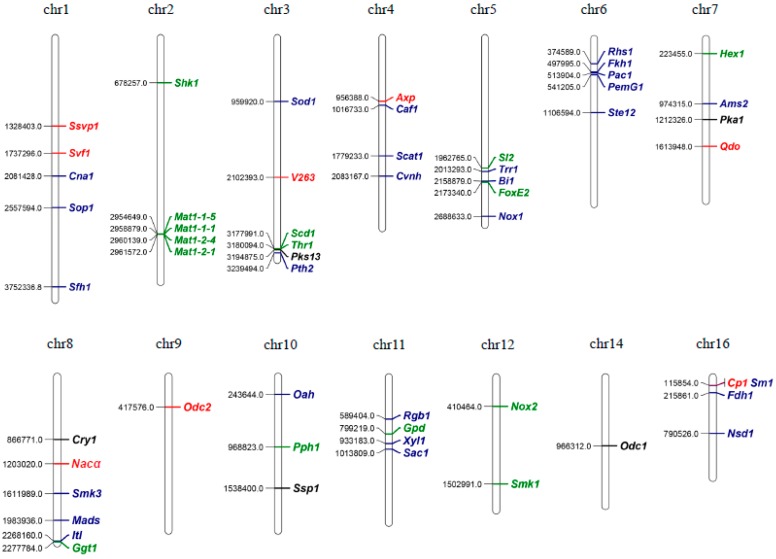
Map positions of the genes in Table 1. The numbers on the left of each chromosome represent the locations of these genes. Genes labelled in green are involved in the development of *S. sclerotiorum*, including hyphal growth, sclerotial formation, apothecial formation and etc. Genes labelled in red mainly play roles in virulence of *S. sclerotiorum*. Genes labelled in blue are involved in both while the ones labelled in black are involved in other biological processes. The chromosomal map was drawn using ‘MapChart’ software.

**Table 1 pathogens-09-00027-t001:** List of all *Sclerotinia sclerotiorum* genes studied so far using mutant analysis. The genes are ordered in regards to their chromosomal positions (Figure 3).

Gene Code (New)	Gene Code (Old)	Strains	Mutant Name	Gene Full Name	Mutant Type	Mutant Phenotypes						Secretion Signal	Tested Hosts	Other Functions	Reference
						Hyphal Growth	Sclerotial Formation	Oxalate Production	Virulence	Appressoria Formation	Induce Host HR/Resistance				
sscle_01g003850	SS1G_02068	Ep-1PNA367, Heilongjiang, China	*ssvp1*	Small secreted virulence-related protein	RNAi	-	NA	NA	+	NA	+	Yes	Canola (Zhongyou 821)	HR induction	[30]
sscle_01g004990	SS1G_01919	1980, Nebraska, USA	*svf1*	Survival factor 1 homologue	RNAi	-	NA	-	+	+	NA	No	Arabidopsis and canola (Zhongyou 821)	Cell wall integrity, ROS production	[31]
sscle_01g006030	SS1G_01788	1980, Nebraska, USA	*cna1*	Catalytic subunit calcineurin-encoding gene	RNAi	+	+	-	+	-	NA	No	Arabiposis and tomato (Bonny Best)	Hyphal elongation	[32]
sscle_01g007450	SS1G_01614	Ep-1PNA367, Heilongjiang, China	*sop1*	Microbial opsin homolog gene	RNAi	+	+	NA	+	NA	NA	No	Arabidopsis	Stress responses	[33]
sscle_01g011030	SS1G_01151	1980, Nebraska, USA	*sfh1*	GATA-box and SNF5 domains containing transcription factor	RNAi	+	+	NA	+	+	NA	No	Soybean, common bean and tomato	ROS accumulation	[34]
sscle_02g013550	SS1G_12694	HA61, Jiangsu, China	*shk1*	Histidine kinases	Deletion	+	+	NA	-	NA	NA	No	Rapeseed, strawberry, tomato and cucumber	Oxidative stresses, glycerol accumulation	[35]
sscle_02g020240	SS1G_04003	1980, Nebraska, USA	*mat1-1-5*	Mating-type gene	Deletion	-	-	NA	-	NA	NA	No	Tomato	Apothecial development	[29]
sscle_02g020250	SS1G_04004	1980, Nebraska, USA	*mat1-1-1*	Mating-type gene	Deletion	-	-	NA	-	NA	NA	No	Tomato	Apothecial development	[29]
sscle_02g020260	SS1G_04005	1980, Nebraska, USA	*mat1-2-4*	Mating-type gene	Deletion	-	-	NA	-	NA	NA	No	Tomato	Apothecial development, ascospore production	[29]
sscle_02g020270	SS1G_04006	1980, Nebraska, USA	*mat1-2-1*	Mating-type gene	Deletion	+	-	NA	-	NA	NA	No	Tomato	Apothecial development	[29]
sscle_03g025030	SS1G_00699	WMA1, Washington, USA	*sod1*	Cu/Zn superoxide dismutase	T-DNA	-	-	-	+	NA	NA	No	Pea (Guido)	Detoxification of host ROS	[36]
sscle_03g025030	SS1G_00699	1980, Nebraska, USA	*sod1*	Cu/Zn superoxide dismutase	Deletion	+	+	+	+	NA	NA	No	Tomato (Garden Peach) and tobacco	Oxidative stress tolerance, repression of host ROS	[37]
sscle_03g028510	SS1G_00263	Isolated from an infected canola stem, Alberta, Canada	*v263*	Hypothetical secreted protein	Deletion	-	NA	NA	+	NA	NA	Yes	Canola		[38]
sscle_03g031470	SS1G_13314	Isolated from diseased rapeseed, Alberta, Canada	*scd1*	Scytalone dehydratase	Deletion	+	+	NA	-	NA	NA	No	Rapeseed (Westar)	Hyphal branching	[39]
sscle_03g031480	SS1G_13315	Isolated from diseased rapeseed, Alberta, Canada	*thr1*	Trihydroxynaphthalene reductase	Deletion	+	+	NA	-	NA	NA	No	Rapeseed (Westar)	Hyphal branching	[39]
sscle_03g031520	SS1G_13322	1980, Nebraska, U.S.A.; UF1, Florida; WMA, Washington, USA	*pks13*	Polyketide synthase	Deletion by CRISPR	-	-	NA	-	+	NA	No	Soybean, canola, tomato (Better Boy), faba bean (Windsor) and pea (Sugar Daddy)	Pigmentation of compound appressoria	[40]
sscle_03g031670	SS1G_13339	274, Nebraska, USA	*pth2*	Peroxysomal carnitine acetyl transferase	Deletion	+	+	+	+	+	NA	No	Soybean	Oxalic acid accumulation	[41]
sscle_04g034810	SS1G_02462	Isolated from an infected canola stem, Alberta, Canada	*axp*	Arabinofuranosidase/β-xylosidase	Deletion	NA	NA	NA	+	NA	NA	Yes	Canola		[42]
sscle_04g034960	SS1G_02486	Sunf-M, Inner Mongolia, China	*caf1*	Secreted protein with a putative Ca2+-binding EF-hand motif	RNAi, T-DNA	+	+	-	+	+	+	Yes	Arabidopsis, rapeseed, pak choi cabbage, hot pepper, tomato, cucumber and soybean		[43]
sscle_04g037170	SS1G_02784	1980, Nebraska, USA	*scat1*	Type A catalase	Deletion	+	+	NA	+	NA	NA	No	Tomato (Garden Peach)	Modulation of ROS	[44]
sscle_04g038020	SS1G_02904	Ep-1PNA367, Heilongjiang, China	*cvnh*	Secreted protein	RNAi	+	+	NA	+	NA	NA	Yes	Arabidopsis		[45]
sscle_05g046240	SS1G_05917	SUN-F-M, Inner Mongolia, China	*sl2*	Cell wall protein	RNAi	-	+	NA	-	NA	NA	Yes	Rapeseed and Arabidopsis	Cellular integrity	[46]
sscle_05g046390	SS1G_05899	1980, Nebraska, USA	*trr1*	Thioredoxin reductase	RNAi	NA	+	NA	+	NA	NA	No	Arabidopsis and tobacco	Oxidative stress tolerance	[47]
sscle_05g046790	SS1G_05839	1980, Nebraska, USA	*bi1*	Bax inhibitor-1 protein	RNAi	+	-	-	+	NA	NA	No	Arabidopsis and tomato	Stress responses, hyphal tip branching	[48]
sscle_05g046830	SS1G_05834	JRUF1, Florida, USA	*foxe2*	Forkhead-box transcription factor family gene	Deletion, T-DNA	-	-	NA	-	-	NA	No	Tomato	Apothecial development	[49]
sscle_05g048220	SS1G_05661	1980, Nebraska, USA	*nox1*	NADPH oxidase	RNAi	-	+	+	+	NA	NA	No	Tomato (Rutger)	ROS regulation	[50]
sscle_06g049430	SS1G_07404	1980, Nebraska, USA	*rhs1*	Rearrangement hot spot repeat-containing protein	RNAi	+	+	-	+	+	NA	Yes	Arabidopsis and canola (Zhongshuang 9)		[51]
sscle_06g049780	SS1G_07360	1980, Nebraska, USA	*fkh1*	Atypical forkhead (FKH)-box-containing protein	RNAi	+	+	NA	+	NA	NA	No	Tomato	Cellular integrity	[52]
sscle_06g049830	SS1G_07355	1980, Nebraska, USA	*pac1*	pH-Responsive transcription factor	Deletion	-	+	+	+	NA	NA	No	Arabidopsis and tomato (Bonnie Best)		[53]
sscle_06g049890	SS1G_07345	NGA4, Anhui, China	*pemg1*	Elicitor-homologous protein	RNAi	+	NA	-	+	+	NA	No	Oilseed rape and soybean	Negative regulator of growth and virulence	[54]
sscle_06g051560	SS1G_07136	UF1, Florida, USA	*ste12*	Downstream transcription factor of MAPK pathway	RNAi	+	+	NA	+	+	NA	No	Bush bean and tomato		[13]
sscle_07g055970	SS1G_03527	SUN-F-M, Inner Mongolia, China	*hex1*	Woronin body major protein	RNAi	-	+	NA	NA	NA	NA	No		Cellular integrity	[46]
sscle_07g058030	SS1G_03252	UF1, Florida, USA	*ams2*	Cell-cycle-regulated GATA transcription factor	RNAi	+	-	NA	+	+	NA	No	Common bean	Chromosome segregation, number and distribution of sclerotia	[55]
sscle_07g058620	SS1G_03171	1980, Nebraska, USA	*pka1*	Protein kinase A	Deletion	-	-	-	NA	NA	NA	No		PKA activity	[56]
sscle_07g059700	MK992913	1980, Nebraska, USA	*qdo*	Quercetin dioxygenase gene	Deletion	-	NA	NA	+	NA	NA	Yes	Arabidopsis	Flavonol degradation	[57]
sscle_08g064670	SS1G_05163	1980, Nebraska, USA	*cry1*	Cryptochrome family CRY-DASH ortholog	Deletion	-	-	NA	-	NA	NA	No	Arabidopsis and tomato	Sclerotial mass, response to UV	[58]
sscle_08g065550	SS1G_05284	NGA4, Anhui, China	*nacα*	Nascent polypeptide-associated complex α-subunit	RNAi	-	-	-	+	NA	NA	No	Oilseed rape and tobacco		[59]
sscle_08g066770	SS1G_05445	1980, Nebraska, USA	*smk3*	Slt2 ortholog	Deletion	+	+	NA	+	+	NA	No	Canola (Westar)	Cuticle penetration, cell wall integrity	[60]
sscle_08g067830	SS1G_05588	1980, Nebraska, U.S.A.	*mads*	MADS-box proteins	RNAi	+	-	NA	+	NA	NA	No	Tomato		[61]
sscle_08g068500	SS1G_14133	Ep-1PNA367, Heilongjiang, China	*itl*	Integrin-like protein	RNAi	+	+	NA	+	NA	+	Yes	Arabidopsis and canola	Hyphal branching, suppression of host defense	[62]
sscle_08g068530	SS1G_14127	1980, Nebraska, USA	*ggt1*	γ-Glutamyl transpeptidase	Deletion	-	+	NA	-	+	NA	Yes	Tomato		[63]
sscle_09g069850	SS1G_10796	1980, Nebraska, USA	*odc2*	Oxalate decarboxylases	Deletion	-	-	+	+	+	NA	Yes	Common bean (Bush Blue Lake 47), soybean (Harosoy), tomato (Bonnie Best) and celery		[64]
sscle_10g075560	SS1G_08218	WMA1, Washington, USA	*oah*	Oxaloacetate acetylhydrolase	T-DNA	-	-	+	+	NA	NA	No	Faba bean (Broad Windsor), pea (Guido), green bean (Great North Tara) and soybean (Skylla)		[65]
sscle_10g075560	SS1G_08218	1980, Nebraska, USA	*oah1*	Oxaloacetate acetylhydrolase	Deletion	-	+	+	+	+	NA	No	Tomato (Bonnie Best), common bean (Bush Blue Lake 47), soybean (Harosoy), canola, Arabidopsis and sunflower		[66]
sscle_10g075560	SS1G_08218	1980, Nebraska, U.S.A.; UF1, Florida; WMA, Washington, USA	*oah1*	Oxaloacetate acetylhydrolase	Deletion by CRISPR	-	+	+	+	+	NA	No	Soybean, canola, tomato (Better Boy), faba bean (Windsor), and pea (Sugar Daddy)		[40]
sscle_10g077630	SS1G_08489	1980, Nebraska, USA	*pph1*	Type 2A Ser/Thr phosphatase catalytic subunit PP2Ac	RNAi	+	+	NA	NA	NA	NA	No			[67]
sscle_10g079320	SS1G_14065	1980, Nebraska, USA	*ssp1*	Development-specific protein	Deletion	-	-	NA	NA	NA	NA	No		Resistance to glycoside-containing antibiotics	[68]
sscle_11g082700	SS1G_07871	1980, Nebraska, USA	*rgb1*	Type 2A Ser/Thr phosphatase B subunit	RNAi	-	+	NA	+	+	NA	No	Arabidopsis and tomato (Bonny Best)	MAPK pathway	[67]
sscle_11g083230	SS1G_07798	SUN-F-M, Inner Mongolia, China	*gpd*	glyceraldehyde-3-phosphate dehydrogenase	RNAi	NA	+	NA	NA	NA	NA	No			[46]
sscle_11g083680	SS1G_07749	1980, Nebraska, USA	*xyl1*	Endo-β-1,4-xylanase	Deletion	+	+	NA	+	NA	NA	Yes	Canola and Arabidopsis	Apothecia formation	[69]
sscle_11g083950	SS1G_07715	1980, Nebraska, USA	*sac1*	Adenylate cyclase	Deletion	+	+	-	+	NA	NA	No	Tomato (Bonnie Best)	Apothecia production, cAMP-signaling	[70]
sscle_12g087830	SS1G_11172	1980, Nebraska, USA	*nox2*	NADPH oxidase	RNAi	-	+	-	-	NA	NA	No	Tomato (Rutger)	ROS regulation	[50]
sscle_12g090900	SS1G_11866	1980, Nebraska, USA	*smk1*	ERK -type MAP kinase	RNAi	+	+	NA	NA	NA	NA	No		pH-dependent regulation	[71]
sscle_14g099710	SS1G_08814	1980, Nebraska, USA	*odc1*	Oxalate decarboxylases	Deletion	-	NA	NA	-	NA	NA	Yes	Common bean (Bush Blue Lake 47), soybean (Harosoy) and tomato (Bonnie Best)		[64]
sscle_16g107670	SS1G_10096	1980, Nebraska, USA	*cp1*	Cerato-platanin protein	Deletion	-	-	NA	+	NA	+	Yes	Arabidopsis and tobacco	Interaction with PR1	[72]
sscle_16g107670	SS1G_10096	FXGD2, Anhui, China	*sm1*	Cerato-platanin protein	RNAi	+	+	-	+	NA	NA	Yes	Tobacco, oilseed rape and soybean		[73]
sscle_16g107930	SS1G_10135	1980, Nebraska, USA	*fdh1*	Formaldehyde dehydrogenase	Deletion	-	+	NA	+	+	NA	No	Common bean and tomato	Osmotic oxidative stress resistance	[74]
sscle_16g109570	ANQ80447	1980, Nebraska, USA	*nsd1*	GATA-type Ivb zinc-finger transcription factor	Deletion	-	-	NA	+	+	NA	No	Tomato (Bonnie Best), celery and tomato	Ascogonia formation, apothecium development	[75]

The assembly genome for the new version gene code is ASM185786v1 published in 2017 and ASM14694v1 for the old version published in 2011. ‘+’ in the table represents that the corresponding phenotype of mutant is altered compared to wild type strain, while ‘-’ means that the phenotype remains unchanged. NA means not assessed. The secretion signals were found using ‘SignalP-5.0’. Unless specified, all deletions were generated with homologous recombination. The Latin names of host species used for pathogenicity test of mutants are: *Apium graveolens* (celery), *Arabidopsis thaliana* (Arabidopsis), *Brassica napus* (canola, oilseed rape, rapeseed), *Brassica rapa* subsp. chinensis (pak choi cabbage), *Capsicum frutescens* (hot pepper), *Cucumis sativus* (cucumber), *Fragaria ananassa* (strawberry), *Glycine max* (soybean), *Helianthus annuus* (sunflower), *Lycopersicon solanum* (tomato), *Nicotiana benthamiana* (tobacco), *Phaseolus vulgaris* (bush bean, common bean, green bean), *Pisum sativum* (pea), *Vicia faba* (faba bean). The ecotype of *A. thaliana* mentioned in the table is Columbia-0. The specific local cultivars of host species are shown in brackets.

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
