# Peer review of "The Notorious Soilborne Pathogenic Fungus Sclerotinia sclerotiorum: An Update on Genes Studied with Mutant Analysis"

_pathogens, 2019, doi:10.3390/pathogens9010027_

Round 1
Reviewer 1 Report
The manuscript entitled “The notorious soilborne pathogenic fungus Sclerotinia sclerotiorum: an update on genes studied with mutant analysis” makes a review of key molecular findings on the biology and pathogenesis process of S. sclerotiorum, focusing on genes that have been studied using mutant analysis. The manuscript is extremely complete, very clear and well written. My only comment is that the abstract is quite reduced, it could include some additional information.
Author Response
We have added more details to the abstract in order to be more comprehensive.
Reviewer 2 Report
This manuscript reviews the information of genes of Sclerotinia sclerotiorum, a pathogenic fungus, and its pathogenic process. However, the reviewer thought that some titles of sections should be changed. Because they don’t indicate contents of them well. Especially, section 2s are more likely to show genomic information than ‘biology’, besides 2.4.
Author Response
We have modified the subtitles to be more accurate.
Reviewer 3 Report
Comments to the authors
The purpose of this work was to provide on update on the key findings of Schlerotinia schlerotiorum based on the newly available molecular data. This fungus is a crop pathogen with a wide range of host species and the ability to persist in the environment, making it challenging to control by chemical or crop management methods. The main purpose of this work was to bring together information on the large number of tested mutants and their effects on fungal growth, schlerotia formation, virulence, HR and other features. This manuscript contained the best-written introduction I have seen in a long time. The background information was presented clearly, with a quality summary of previous work, and proper citations. The sections delving into detail on each mutant are a bit dense, as would be expected. A few points of clarification would help to strengthen the manuscript.
Major consideration
This fungus had a very broad host range. It is likely that the general mechanism used by the fungus to successfully colonize plants is shared between hosts. However, in case there are host-specific differences, it would be very useful to know which plant species were used to test virulence and HR. Please add these information.
How variable are the listed genes (table 1) between the various strains of Schlerotinia schlerotiorum? Do they show regional variation or strain to strain differences? Was the same strain used for all the experiments listed in Table 1?
This reviewer strongly suggests not using red and green and informative colors in Figure 1. These cannot be differentiated by a substantial portion of the population. Also, when printed in greyscale the colors are nearly identical for most gene names. Please consider using underlines, bold or other means to help the reader tell them apart.
A few minor considerations
Please add the scientific names of the crop species as well as their common names.
Table 1 needs some formatting to ensure that gene codes and title headings do not have words split into two lines. It is a very large table with lots of information. If it does end up split onto multiple pages please ensure that the column headings are retained on each page.
The authors may consider adding a figure diagraming key stages of fungal growth and pathogenesis, and illustrate where genes (or gene categories) function during these processes.
Author Response
This fungus had a very broad host range. It is likely that the general mechanism used by the fungus to successfully colonize plants is shared between hosts. However, in case there are host-specific differences, it would be very useful to know which plant species were used to test virulence and HR. Please add these information.
This is a great suggestion. We have now added the host information in Table 1.
How variable are the listed genes (table 1) between the various strains of Schlerotinia schlerotiorum? Do they show regional variation or strain to strain differences? Was the same strain used for all the experiments listed in Table 1?
To clarify, we also added the specific strain information used in each study in Table 1.
This reviewer strongly suggests not using red and green and informative colors in Figure 1. These cannot be differentiated by a substantial portion of the population. Also, when printed in greyscale the colors are nearly identical for most gene names. Please consider using underlines, bold or other means to help the reader tell them apart.
This is a valid point. We have modified Fig 1 to a black and white design.
A few minor considerations
Please add the scientific names of the crop species as well as their common names.
We have added the host species and common names in Table 1 and appropriate mentions.
Table 1 needs some formatting to ensure that gene codes and title headings do not have words split into two lines. It is a very large table with lots of information. If it does end up split onto multiple pages please ensure that the column headings are retained on each page.
We have paid attention to the format of Table 1 so it looks more consistent.
The authors may consider adding a figure diagraming key stages of fungal growth and pathogenesis, and illustrate where genes (or gene categories) function during these processes.
A new figure is added as suggested (current Fig 1).
